# A Selective Etching Route for Large-Scale Fabrication of β-Ga_2_O_3_ Micro-/Nanotube Arrays

**DOI:** 10.3390/nano11123327

**Published:** 2021-12-07

**Authors:** Shan Ding, Liying Zhang, Yuewen Li, Xiangqian Xiu, Zili Xie, Tao Tao, Bin Liu, Peng Chen, Rong Zhang, Youdou Zheng

**Affiliations:** Key Laboratory of Advanced Photonic and Electronic Materials, School of Electronic Science and Engineering, Nanjing University, Nanjing 210023, China; dg20230010@smail.nju.edu.cn (S.D.); dg1723035@smail.nju.edu.cn (L.Z.); 141180062@smail.nju.edu.cn (Y.L.); zlxie@nju.edu.cn (Z.X.); ttao@nju.edu.cn (T.T.); bliu@nju.edu.cn (B.L.); pchen@nju.edu.cn (P.C.); ydzheng@nju.edu.cn (Y.Z.)

**Keywords:** β-Ga_2_O_3_, micro-/nanotubes, GaN template, ICP etching

## Abstract

In this paper, based on the different etching characteristics between GaN and Ga_2_O_3_, large-scale and vertically aligned β-Ga_2_O_3_ nanotube (NT) and microtube (MT) arrays were fabricated on the GaN template by a facile and feasible selective etching method. GaN micro-/nanowire arrays were prepared first by inductively coupled plasma (ICP) etching using self-organized or patterning nickel masks as the etching masks, and then the Ga_2_O_3_ shell layer converted from GaN was formed by thermal oxidation, resulting in GaN@Ga_2_O_3_ micro-/nanowire arrays. After the GaN core of GaN@Ga_2_O_3_ micro-/nanowire arrays was removed by ICP etching, hollow Ga_2_O_3_ tubes were obtained successfully. The micro-/nanotubes have uniform morphology and controllable size, and the wall thickness can also be controlled with the thermal oxidation conditions. These vertical β-Ga_2_O_3_ micro-/nanotube arrays could be used as new materials for novel optoelectronic devices.

## 1. Introduction

The monoclinic β-Ga_2_O_3_ has become one of the most important functional materials for high-power applications and UV detection. It has a wide bandgap (~4.9 eV at room temperature), a high expected breakdown electric field (8 MV/cm), great thermal and chemical stability. In recent years, β-Ga_2_O_3_ micro-/nanostructures have exhibited technological potential in many device applications, such as field-effect transistors [1,2,3,4], photodetectors [5,6,7], gas sensors [8,9,10], solar cells [11], and nanophotonic switches [12].

So far, the research on the tubular structure of β-Ga_2_O_3_ is very limited, and only several methods have been developed for synthesizing β-Ga_2_O_3_ micro-/nanotubes [13,14,15,16,17]. Cheng et al. [13] firstly fabricated vertical β-Ga_2_O_3_ nanotubes (NTs) in a template-based sol-gel method, the size of the NTs can be adjusted by changing the template dimensions and the sol immersion time. Kong et al. [15] synthesized β-Ga_2_O_3_ NTs on an Au-coated silicon substrate by physical evaporation. Braniste et al. [17] reported the growth of microtubular aero-GaN on ZnO sacrificial templates, and then oxidized the GaN to fabricate interconnected β-Ga_2_O_3_ microtubes (MTs). Although efforts have been made to fabricate β-Ga_2_O_3_ micro-/nanotube, the β-Ga_2_O_3_ micro-/nanotube reported were mostly disordered or inclined. In addition, the preparation process of the above methods is complicated, it is difficult to control the size and wall thickness accurately during the growth of tubes. Therefore, the synthesis of large-scale growth of β-Ga_2_O_3_ tubular structure arrays with uniform morphology is still a huge challenge. At present, the inductively coupled plasma (ICP) etching technology for the preparation of GaN nanowire (NW) arrays is mature, but there are few reports on the preparation of β-Ga_2_O_3_ NW arrays by ICP etching. The existing reports mainly study the etching process conditions of β-Ga_2_O_3_ films and single crystal [18,19,20]. The etching behavior of Ga_2_O_3_ is completely different from that of GaN, more similar to Al_2_O_3_ [21]. Our group conducted a systematic study on the ICP etching characteristics of β-Ga_2_O_3_ and the preparation process of β-Ga_2_O_3_ NW structure, and successfully prepared GaN and β-Ga_2_O_3_ NW arrays by ICP etching technology based on Cl-based plasma, respectively [22,23]. These studies indicate that Ga_2_O_3_ requires relatively higher etching power and etching gas flow than GaN. Moreover, a uniform β-Ga_2_O_3_ shell would be formed on the GaN surface under suitable oxidation conditions [24,25,26], the thickness and crystallinity of the shell layer could be controlled by varying the oxygen pressure, oxidation temperature, and oxidation time. Based on this, the vertical GaN@Ga_2_O_3_ core-shell heterostructured NW arrays were prepared by thermal oxidation of GaN NW arrays [27].

Motivated by the above study, herein, we propose a new method based on selective dry etching for large-scale manufacturing of vertically aligned β-Ga_2_O_3_ NT array, that is, the Ga_2_O_3_ NT arrays can be fabricated by etching away the GaN core of GaN@Ga_2_O_3_ core-shell NW arrays. Besides, a periodic array of β-Ga_2_O_3_ MTs was fabricated through the same route based on GaN microwires (MWs) prepared by photolithography technology. Compared with the traditional bottom-up synthesis method, this route is easy to control and suitable for large-scale preparation. In this study, the formation mechanism of the prepared β-Ga_2_O_3_ tubular structure was discussed. Meanwhile, the influence of oxidation time and temperature on the tubular structure was investigated.

## 2. Materials and Methods

### 2.1. Preparation of GaN Micro-/Nanowire Arrays

A 4 μm-thick GaN epitaxial film used in this study was grown on c-plane sapphire substrates by metal organic chemical vapor deposition (MOCVD) [28], and a 200 nm-thick SiO_2_ protective layer was deposited by plasma enhanced chemical vapor deposition (PECVD) on the GaN epitaxial wafer. The GaN micro-/nanowire arrays were fabricated by the top-down ICP etching process based on Ni masks. The GaN NW arrays were prepared by etching the GaN template through a self-assembled Ni nanomask formed by rapid thermal annealing (RTA) [22]. Meanwhile, the well-aligned GaN MW arrays were realized by using a micropatterning process and an etching process. For the photolithography process, the S1805 photoresist and a photomask with a hexagonal shape diameter of 2 μm were used to form hole arrays on the GaN epitaxial layer deposited with the SiO_2_ layer. After a Ni layer was evaporated using electron beam evaporation (EBE) and followed by lift-off operation, periodic Ni island arrays with a diameter of about 1.7 μm were obtained, serving as a dry etching mask.

For dry etching, the GaN template was cut into a 1 × 1 cm^2^ square sample. After forming the etching mask on the surface of the GaN epitaxial film, the sample was placed on the Si substrate holder of the ICP system for etching. In the ICP (Oxford Plasmalab System 100) system, the plasma was generated by a radio frequency (13.56 MHz) glow discharge, the He backside cooled chuck (4 inch diameter) was biased with 13.56 MHz RF power to control ion energy. The temperature of the substrate holder was kept at 20 °C by using a water-flow cooling system. The inlet flow is adjusted by mass flow controllers, the outlet flow is controlled by the butterfly valve module. The vacuum chamber is exhausted by a molecular pump. The optimized ICP etching conditions for GaN were performed from our previous work [22,29,30], the 90%Cl_2_/10%BCl_3_ (48/6 sccm) was used with the RF/ICP power fixed at 100/300 W, while the DC bias voltage generated was 460 V, the chamber pressure and helium backing were kept at 10 mTorr and 6 Torr, and the sample was etched for 5 min.

### 2.2. Fabrication of β-Ga_2_O_3_ Micro-/Nanotube Arrays

Figure 1 schematically depicts the preparation steps of β-Ga_2_O_3_ micro-/nanotubes in this study. GaN micro-/nanowire arrays were placed in a quartz tube for thermal oxidation to form GaN@Ga_2_O_3_ micro-/nanowire arrays [24,27]. During the oxidation process, the pure oxygen flow was maintained at 200 sccm, and the oxidation temperature was fixed at 850 °C, 900 °C and 950 °C. Then the vertical β-Ga_2_O_3_ micro-/nanotube arrays were fabricated by selective etching of the GaN@Ga_2_O_3_ micro-/nanowire arrays using the ICP etching technique. Before etching, the prepared GaN@Ga_2_O_3_ micro-/nanowire arrays were immersed in a buffer oxide etching (BOE) solution to remove the remaining Ni/SiO_2_ mask layer. GaN@Ga_2_O_3_ micro-/nanowire arrays were etched using ICP etching conditions for GaN to prepare β-Ga_2_O_3_ micro-/nanotube arrays. The Cl_2_/BCl_3_ (48/6 sccm) was used as the etching gas, RF/ICP power was fixed at 100/300 W, and the chamber pressure was kept at 10 mTorr for 5 min of etching time.

### 2.3. Characterizations

The surface morphology, size distribution, and chemical composition of the prepared samples were characterized by a scanning electron microscope (SEM, GeminiSEM 500) with energy dispersive spectroscopy (EDS). A high-resolution transmission electron microscope (HRTEM, Tecnai G^2^ F20 S-TWIN) was used to characterize the crystallographic and structural properties of β-Ga_2_O_3_ NTs samples. The preparation of TEM samples is a simple process. The NTs were first scraped from the substrate surface and dispersed in ethanol by ultrasonic agitations. Then a few droplets of the suspension were placed on a copper grid covered with a perforated carbon film and finally dried at room temperature in ambient air.

## 3. Results and Discussion

The vertically aligned GaN and β-Ga_2_O_3_ NW arrays were successfully fabricated by ICP etching using self-assembled Ni nanoislands as the etching masks [23]. ICP power/RF power, chamber pressure, and Cl_2_/BCl_3_ gas mixing ratio were adjusted to investigate the effect of input process parameters on the etch characteristics of GaN and β-Ga_2_O_3_ films. The etching rates of the prepared GaN and β-Ga_2_O_3_ NWs under optimized etching conditions are shown in Table 1. The results show that ICP etching of β-Ga_2_O_3_ requires higher gas flow, higher etching power, and higher chamber pressure, but the achieved etching rate is still very low, which means that β-Ga_2_O_3_ would not be etched away completely while GaN is etched rapidly under the same etching conditions.

Figure 2 shows a schematic diagram of the formation mechanism of β-Ga_2_O_3_ micro-/nanotubes based on the large etching difference between GaN and β-Ga_2_O_3_ [21,31]. The GaN@Ga_2_O_3_ micro-/nanowires are etched using the ICP etching condition for GaN to remove the GaN core while retaining the Ga_2_O_3_ shell, to prepare Ga_2_O_3_ micro-/nanotubes with a hollow structure. On the one hand, the main etching component for GaN is Cl_2_ of Cl_2_/BCl_3_ ICP-plasmas. For GaN, the Cl radicals in the Cl_2_ plasma are the main chemical reactant, which can chemically react with GaN to obtain the volatile etch product GaCl_3_. BCl_3_ can act as a wetting agent, and the BCl^+3^ ions and BCl^+2^ ions are mainly used to remove GaCl_3_ by physical sputtering [32,33]. However, for Ga_2_O_3_, BCl_3_ plasma exhibits a stronger etching effect, and Cl_2_ plasma exhibits a weaker etching effect [21]. BCl^+2^ ions react chemically with Ga_2_O_3_. B_3_Cl_3_O_3_ gas is produced when the oxygen pressure is lower than the critical oxygen pressure. When the oxygen pressure is higher than the critical oxygen pressure, a nonvolatile B_2_O_3_ product will be formed. The product will accumulate and form a barrier layer on the Ga_2_O_3_ surface, hindering the further etching of Ga_2_O_3_ [19,34]. On the other hand, Ga_2_O_3_ is a strongly bonded material [35], and its ICP etching requires higher gas flow and etching power. Therefore, under the conditions of ICP etching GaN, β-Ga_2_O_3_ will not be effectively etched, thus forming the β-Ga_2_O_3_ micro-/nanotubes.

Figure 3a,d shows the SEM images of vertical GaN NW arrays on the sapphire substrate. The density of the GaN NWs is about 8.7 × 10^8^ cm^−2^, and the average height and diameter are 1.6 μm and 200 nm, respectively. From Figure 3d, it can be observed that the remaining Ni/SiO_2_ mask layer is on the top of the GaN NWs with relatively vertical sidewall profiles. Compared with other soft masks, this mask has better anisotropic etching characteristics for GaN materials. Besides, to a certain degree, the mask layer can prevent the reaction and diffusion of O_2_ from the top of the NW in the subsequent thermal oxidation process.

The GaN NW array sample was thermally oxidized for 10 min in an environment with an oxygen flow of 200 sccm and a temperature of 950 °C. This process can realize the transformation of crystalline structures from GaN NWs into GaN@Ga_2_O_3_ NWs. The SEM images of GaN@Ga_2_O_3_ NWs after BOE treatment are shown in Figure 3b,e. It is observed from the figures that the NW array maintains its original shape, but its surface becomes rough after the thermal oxidation. In addition, the uniform Ga_2_O_3_ shell layer was formed on the surface of GaN NWs with an average thickness of about 60 nm (Figure 3b). The GaN@Ga_2_O_3_ NW arrays were etched under the etching process conditions for GaN (Table 1), and β-Ga_2_O_3_ NT arrays were obtained, as shown in Figure 3c,f. The inset of Figure 3c shows a magnified tilted-view SEM image of β-Ga_2_O_3_ NTs, showing a hollow internal structure, which indicates that the ICP process effectively etched the GaN core.

The grazing incidence in-plane geometry was used to determine the crystal structure of Ga_2_O_3_ grains. Figure 4 shows the in-plane diffraction patterns of unoxidized GaN nanowires and Ga_2_O_3_ nanotubes oxidized at 950 °C for 5 min, 10 min, and 15 min. The XRD reflections at 18.9°, 30.2°, 31.7°, 35.1°, 37.4°, 60.0°, and 64.7° were assigned to the beta phase of monoclinic Ga_2_O_3_ planes (−201), (400), (002), (111), (401), (−603) and (−712), respectively. The result indicates that GaN is converted to polycrystalline β-Ga_2_O_3_ after thermal oxidation treatment.

Figure 5a shows the TEM image of β-Ga_2_O_3_ NT oxidized at 950 °C for 10 min. As expected, the cavities of the NT can be clearly seen. It can also identify a clear interface on the inner wall of the tube. The outer diameter and wall thickness of the NT is about 200 and 60 nm, respectively. The selected area electron diffraction (SAED) image is shown in Figure 5b, and the diffraction pattern indicates that the β-Ga_2_O_3_ NT is polycrystalline. From the HRTEM image in Figure 5c, the lattice spacing of approximately 0.282 nm corresponds to (002) plane of β-Ga_2_O_3_ [36]. In addition, the EDS element mapping and the line-scan images of individual NT were also performed. The yellow, red, and blue images of the EDS mapping analysis in Figure 5e illustrate the presence of Ga, N, and O elements, respectively. It can be seen that Ga and O elements are mainly distributed on the NT, while the distribution of N elements has no obvious aggregation. Figure 5f shows the EDS line-scan analysis of the marked red line area in Figure 5d. It can be seen that the peak profile of Ga and O elements is valley-shaped, the intensity on both sides is higher than the intensity at the center, and the thickness of the tube wall is about 60 nm. The content of N element is very low. These results are consistent with the TEM image and prove a hollow tubular structure.

To further study the relationship between the oxidation parameters and the structure of β-Ga_2_O_3_ NTs, the GaN NWs were oxidized at different times and temperatures, and the oxygen flow rate was fixed at 200 sccm. Figure 6a–c shows the SEM images of β-Ga_2_O_3_ NT arrays prepared by oxidation at a temperature of 950 °C for 5 min, 10 min, and 15 min. It can be seen that the wall thickness of the NTs is different under different oxidation times, and the oxidation reaction of GaN starts from the surface of the NW and then gradually diffuses into the interior. Figure 6d,e shows the SEM images of β-Ga_2_O_3_ NT arrays prepared by oxidation at 850 °C and 900 °C for 15 min. When the oxidation temperature increases from 850 °C to 900 °C, the wall thickness of the NTs increases slightly. The wall thickness of the NTs increases sharply as the oxidation temperature rises to 950 °C (Figure 6c). Figure 6f shows the oxidation rate of β-Ga_2_O_3_ NTs under different oxidation conditions. The red line represents the oxidation rate of β-Ga_2_O_3_ NTs oxidized for 15 min at 850 °C, 900 °C, and 950 °C. The oxidation rate vs. temperature relationships yield linear Arrhenius plots [37], indicating that the oxidation process in the regime is under interfacial reaction-controlled. The black line represents the oxidation rate of β-Ga_2_O_3_ NTs oxidized under different oxidation times at 950 °C. The oxidation rate decreases gradually as the oxidation time increases, which means that the oxidation process may be limited by a diffusion-controlled mechanism [38,39]. It could be seen from the above experimental results that Ga_2_O_3_ NT arrays with different inner diameter wall thicknesses can be synthesized by adjusting the oxidation parameters.

The GaN NWs etched by self-assembled Ni metal mask technology are usually unevenly distributed [40], resulting in uneven distribution of subsequent NTs. Therefore, the GaN templates were patterned by photolithography, and then periodic MT arrays with an outer diameter of about 1.7 μm were prepared using the same route. The β-Ga_2_O_3_ MT arrays prepared by oxidation for 20 min, 30 min, and 40 min in an environment at a temperature of 950 °C are shown in Figure 7. The MTs also have a hollow structure with a density of about 7 × 10^6^ cm^−2^. As the oxidation time increases, the corresponding wall thicknesses are about 130 nm, 150 nm, and 155 nm, respectively (Figure 7a–c). In addition, more needle-like nanostructures appeared on the GaN surface (Figure 7d–f). GaN needle-like nanostructures tend to be formed at the defects of the GaN template. The reason is that the GaO_x_ products migrate on the surface and accumulate to form large grains at the surface defects during the thermal oxidation [25], which serves as a mask to form needle-like nanostructures on the GaN surface during the etching process.

The above results prove the feasibility and versatility of this method for preparing vertical well-ordered tubular arrays. The periodic Ga_2_O_3_ NT arrays would be obtained by nanopatterning using e-beam or nanoimprint lithographic techniques. The method is novel and relatively simple, which can realize the control of the size, length, and wall thickness of the β-Ga_2_O_3_ tubular structure, providing possibilities for applications in catalytic, gas-sensing, optic, electronic, photochemical, and terahertz communications [17,41,42,43,44]. The hybrid micro-/nanoLEDs with high-performance red/green/blue and white emissions have been demonstrated in our previous work [45,46]. The preparation of white light LEDs integrated with β-Ga_2_O_3_ micro-/nanotubes through photolithography and selective etching route is now undergoing. A schematic of the LED device is shown in Figure 8. β-Ga_2_O_3_ micro-/nanotube arrays on the surface of InGaN/GaN MQW epitaxial wafers are prepared through a selective etching route, which replaces the nanoholes previously prepared by nanoimprint lithography technique [45]. Then the red and green quantum dots are filled into the micro-/nanotubes to blend white light sources.

## 4. Conclusions

In summary, we developed a simple selective dry etching approach to fabricate the large-scale β-Ga_2_O_3_ micro-/nanotube arrays with controllable dimensions on the GaN template. The wall thickness of the Ga_2_O_3_ tube is related to the oxidation parameters and varies with the oxidation temperature and time. Through this route, GaN@Ga_2_O_3_ micro-/nanowires with different oxide shell thicknesses were prepared by changing thermal oxidation parameters, and β-Ga_2_O_3_ micro-/nanotubes were obtained under selective dry etching. It is expected that the β-Ga_2_O_3_ micro-/nanotube arrays would be used as new materials for novel applications in optoelectronics, such as photodetectors, gas sensors, and electroluminescent devices.

## Figures and Tables

**Figure 1 nanomaterials-11-03327-f001:**
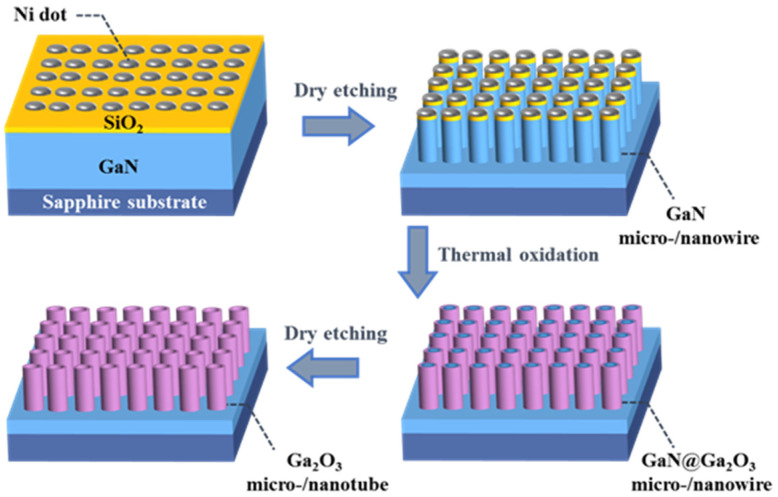
Schematic illustration for the synthesis β-Ga_2_O_3_ micro/nanoarrays.

**Figure 2 nanomaterials-11-03327-f002:**
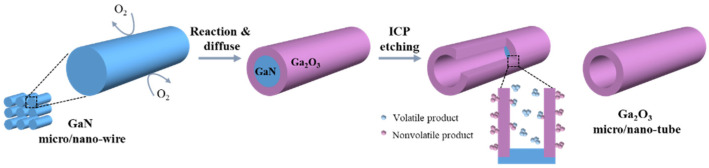
Schematic illustration of the formation process of β-Ga_2_O_3_ micro-/nanotube depicted as a selective etching between GaN and β-Ga_2_O_3_.

**Figure 3 nanomaterials-11-03327-f003:**
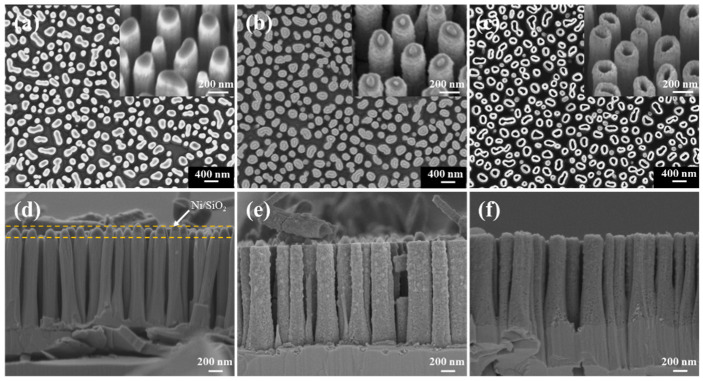
SEM images (**a**,**d**) of the GaN nanowire (NW) arrays before BOE treatment. SEM images (**b**,**e**) of the GaN@Ga_2_O_3_ NW arrays oxidized at 950 °C for 10 min. SEM images (**c**,**f**) of the β-Ga_2_O_3_ NT arrays after ICP etching. The inset of (**a**–**c**) is the tilted-view SEM image.

**Figure 4 nanomaterials-11-03327-f004:**
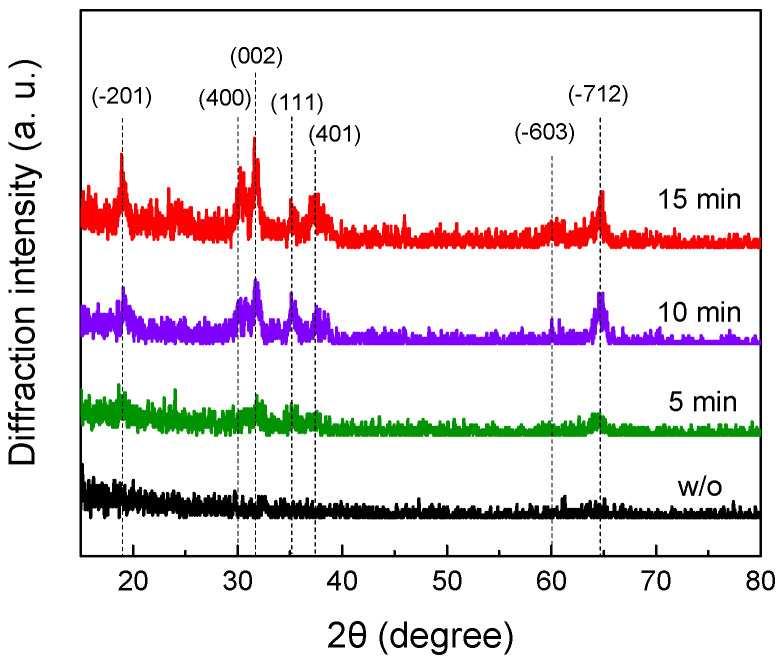
Grazing incidence in-plane XRD analysis of β-Ga_2_O_3_ NTs oxidized at 950 °C and GaN NWs (w/o). Oxidation times and peak assignment are indicated in the figure.

**Figure 5 nanomaterials-11-03327-f005:**
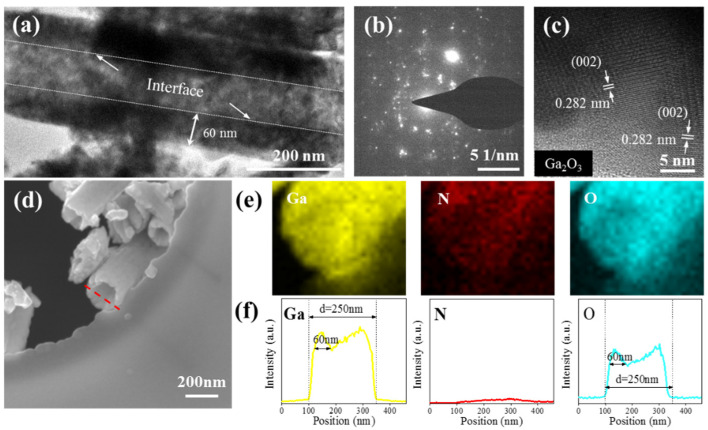
(**a**) TEM image of a single β-Ga_2_O_3_ nanotube (NT) oxidized at 950 °C for 10 min. (**b**) SAED pattern and (**c**) HRTEM image of the β-Ga_2_O_3_ NT. (**d**) The SEM graph of individual Ga_2_O_3_ NT, (**e**) corresponding EDS elemental mapping images, and (**f**) EDS line scanning.

**Figure 6 nanomaterials-11-03327-f006:**
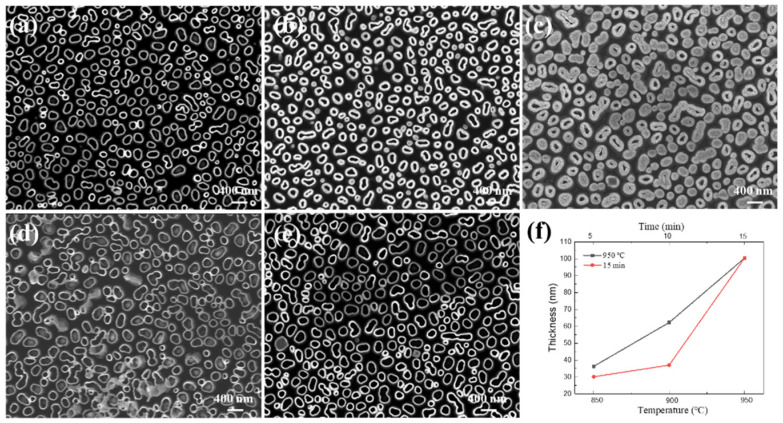
SEM images of the β-Ga_2_O_3_ NT arrays prepared in different oxidation times at 950 °C for (**a**) 5 min, (**b**) 10 min, (**c**) 15 min. SEM images of the β-Ga_2_O_3_ NT arrays prepared in different oxidation temperatures for 15 min at (**d**) 850 °C and (**e**) 900 °C. The oxidation rate images (**f**) of β-Ga_2_O_3_ NTs under different oxidation conditions.

**Figure 7 nanomaterials-11-03327-f007:**
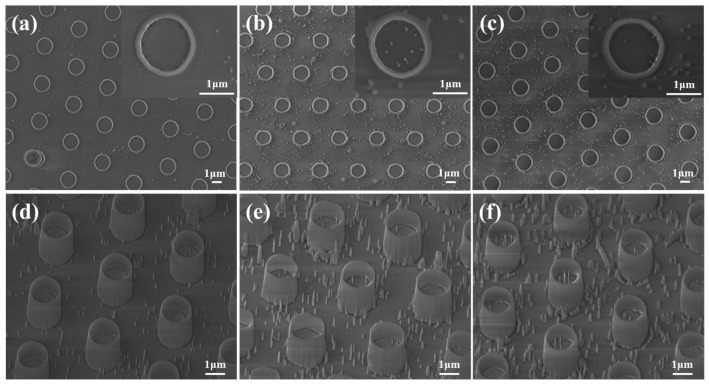
Plane and tilted view SEM images of the β-Ga_2_O_3_ microtubes (MTs) samples oxidized at 950 °C for (**a**,**d**) 20 min, (**b**,**e**) 30 min, and (**c**,**f**) 40 min. Insets at the top right corner show high-magnification images of the same sample.

**Figure 8 nanomaterials-11-03327-f008:**
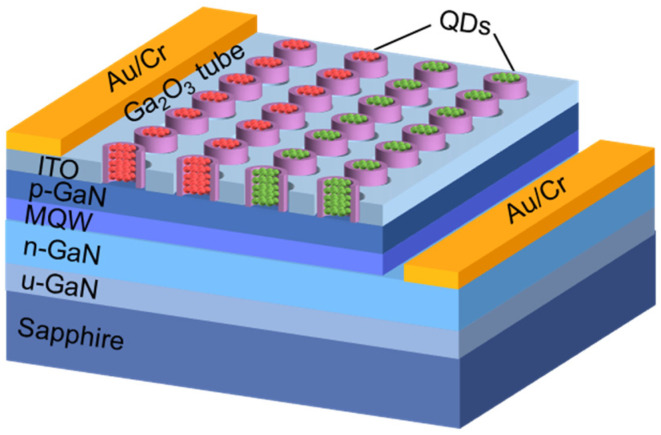
Schematic of the white LED with β-Ga_2_O_3_ micro-/nanotubes.

**Table 1 nanomaterials-11-03327-t001:** ICP etch conditions studied for GaN and β-Ga_2_O_3_ nanowires.

Sample	Cl_2_/BCl_3_ (sccm)	RF Power (W)	ICP Power (W)	Pressure (mTorr)	Etch Rate (nm/min)
GaN	48/6	100	300	10	320
β-Ga_2_O_3_	20/80	150	700	20	50

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
