# Peer review of "A Selective Etching Route for Large-Scale Fabrication of β-Ga2O3 Micro-/Nanotube Arrays"

_nanomaterials, 2021, doi:10.3390/nano11123327_

Round 1

Reviewer 1 Report

In this paper, the authors reported new fabrication process of β-Ga2O3 micro/nano-tube arrays by a selective etching (ICP process). Also, the manuscript is organized systematically and reasonably based on the detailed analysis. But, However, this method does not seem to suggest new concepts. For the above reasons, I can’t recommend publishing the present manuscript in Nanomaterials. Or, i would like to recommend its publication in Nanomaterials after the major revision.

There are many reports on how to fabricate Ga2O3 nanowires. In many papers, Ga2O3 nanowires with a diameter of several tens of nanometers to several hundred nanometers in diameter were fabricated by various methods such as synthesis and growth methods and applied to various devices.

In this method, Ga2O3 layer was produced on the surface of the GaN wire through an oxidation method, and then a Ga2O3 tube was produced through selective etching of GaN. Also, the manuscript is organized systematically and reasonably based on the detailed analysis. But, However, this manuscript only introduces the process for fabrication of Ga2O3 nanotubes. Currently, manuscripts for many studies not only introduce a new process, but also show the results of the properties of materials or devices produced by the process to support the excellence of the process. In order to introduce the superiority of this process developed by this research group, it is necessary to present the results showing what kind of characteristic advantages the produced Ga2O3 nanotubes have compared to the conventoinal Ga2O3 nanowire or Ga2O3 film.

Although the title of this manuscript states that it is a selective etching of GaN, it has already been reported that the etching rate of GaN is faster than that of Ga2O3 when ICP etching is used. It is judged that the results of improving the properties of Ga2O3 fabricated by this method and the application results of devices are necessary.

In addition, further analysis is needed to show that Ga2O3 crystals are well formed. Although the EDX data in Figure 4 and SAED of TEM are shown, Ga2O3 crystals suggest adding XRD analysis results.

Reviewer 2 Report

see attached file

Round 2

Reviewer 2 Report

In my first review, I have urged the authors to define and describe the exact conditions of this RIE process to demonstrate that they make science, not black magic. 

More or less, they failed to do that. Only marginal comments for ICP power and gas mixing ratio, but just one value in each case. 

In the RIE process:

  • No residual gas vacuum,
  • no remarks how to control the residual gas vacuum and the process pressure,
  • no description of process control (temperature control of the substrate, He backside cooling?, control by applying OES or mass spectrometry?), nothing is said about plasma control via impedance spectroscopy (using a Z-scan) or self-excited electron resonance spectroscopy (SEERS, using a Hercules sensor)
  •  simply no communication concerning the dependence of the parameters etch rate and anisotropy on plasma parameters (pressure, gas mixing ratio, ICP power, RF power, DC bias).
  • What is with radial uniformity of the etching process? What was the size of the substrates? How many substrates were simultaneously subjected to etching? 

In the oxidation process:

I proposed two possible tracks to discriminate between the two principle mechanisms of oxidation following their plot in Fig. 6f. Is the oxidation diffusion controlled or does it show the behavior of a thermally-activated Arrhenius law? 

Conclusion:

The paper is the communication of an optimization process without communicating the dependence of their parameters etch rate and anisotropy on plasma parameters. Moreover, not only one application is shown.  

To avoid rejection of their manuscript, either the authors

  • demonstrate an application of their process during preparation of an electronic device or they
  • describe the process following scientific principles. 
